# Hyperrealistic neural decoding: Reconstruction of face stimuli from fMRI measurements via the GAN latent space

## Abstract

We introduce a new framework for hyperrealistic reconstruction of perceived naturalistic stimuli from brain recordings. To this end, we embrace the use of generative adversarial networks (GANs) at the earliest step of our neural decoding pipeline by acquiring functional magnetic resonance imaging data as subjects perceived face images created by the generator network of a GAN. Subsequently, we used a decoding approach to predict the latent state of the GAN from brain data. Hence, latent representations for stimulus (re-)generation are obtained, leading to state-of-the-art image reconstructions. Altogether, we have developed a highly promising approach for decoding sensory perception from brain activity and systematically analyzing neural information processing in the human brain.

## 1 Introduction

In recent years, the field of neural decoding has been gaining more and more traction as advanced computational methods became increasingly available for application on neural data. This is a very welcome development in both neuroscience and neurotechnology since reading neural information will not only help understand and explain human brain function but also find applications in brain computer interfaces and neuroprosthetics to help people with disabilities.

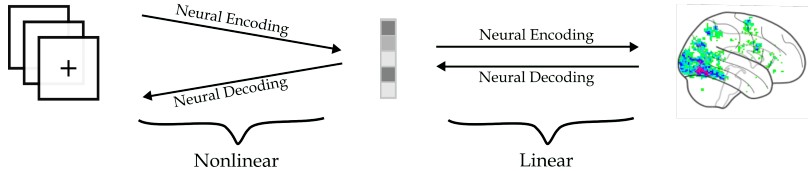

Figure 1: The mapping between sensory stimuli (left) and fMRI recordings (right). Neural encoding seeks to find a transformation from stimulus to the observed brain response via a latent representation (middle). Conversely, neural decoding seeks to find the information present in the observed brain responses by a mapping from brain activity back to the original stimulus.

Neural decoding can be conceptualized as the inverse problem of mapping brain responses back to sensory stimuli via a latent space (20). Such a mapping can be idealized as a composite function of linear and nonlinear transformations (Figure 1). The linear transformation models the mapping from brain responses to the latent space. The latent space should effectively capture the defining properties of the underlying neural representations. The nonlinear transformation models the mapping from the latent space to sensory stimuli.

The systematic correspondences between latent representations of discriminative convnets and neural representations of sensory cortices are well established (23; 14; 2; 7; 8; 6). As such, exploiting these systematic correspondences in neural decoding of visual experience has pushed the state-of-the-art forward (20). This includes linear reconstruction of perceived handwritten characters (15), neural decoding of perceived and imagined object categories (10), and reconstruction of natural images (17; 16) and faces (9; 21). Yet, there is still much room for improvement since state-of-the-art results still fall short of providing photorealistic reconstructions.

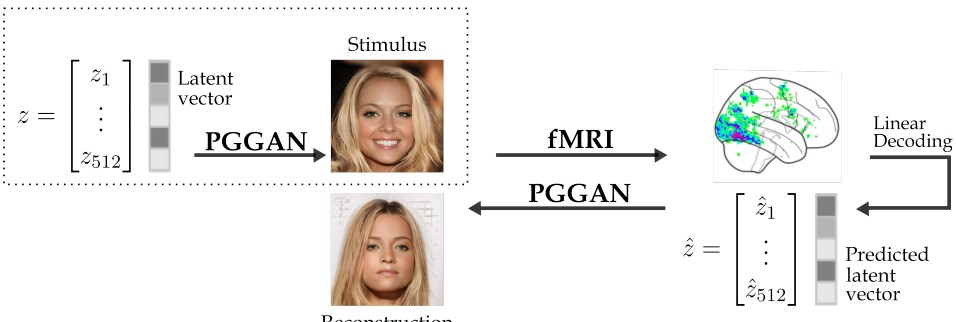

Figure 2: Schematic illustration of the HYPER framework. Face images are generated from randomly sampled latent features $z \in Z$ by a face-generating GAN, as denoted by the dotted box. These faces are then presented as visual stimuli during brain scanning. Next, a linear decoding model learns the mapping from brain responses to the original latent representation, after which it predicts latent features $\hat{z}$ for unseen brain responses. Ultimately, these predicted latent features are fed to the GAN for image reconstruction.

At the same time, generative adversarial networks (GANs) have emerged as perhaps the most powerful generative models to date (5; 11; 12; 1) that can potentially bring neural decoding to the next level. However, since the *true* latent representations of GANs are not readily available for preexisting neural data (unlike those of the aforementioned discriminative convnets), the adoption of GANs in neural decoding has been relatively slow (see (16) for an earlier attempt with GANs and (21) for a related attempt with VAEs).

In this study, we introduce a very powerful yet simple framework for HYperrealistic reconstruction of PERception (HYPER), which elegantly integrates GANs in neural decoding by combining the following components (Figure 2):

i **GAN**. We used a pretrained GAN, which allows for the generation of meaningful data samples from randomly sampled latent vectors. This model is used both for generating the stimulus set and for the ultimate reconstruction of perceived stimuli. In the current study, we used the progressive growing of GANs (PGGAN) model (11), which generates photorealistic faces that resemble celebrities.

ii **fMRI**. We made use of neural data with a known latent representation, obtained by presenting the stimulus set produced using the above-mentioned generative model, and recording the brain responses to these stimuli. In the current study, we collected fMRI recordings in response to the images produced using the PGGAN. We created a dataset consisting of a separate training and test set.

iii **Decoding model**. We used a decoding model, mapping the neural data to the latent space of the generative model. Using this model, we then obtained latent vectors for the neural responses corresponding to the stimulus images in the test set. Feeding these latent vectors back into the generative model resulted in the hyperrealistic reconstructions of perception.

## 2 Methods

### 2.1 Training on synthetic images with known latent features

State-of-the art face reconstruction techniques use deep neural networks to encode vectors of latent features for the images presented during the fMRI experiment (9; 21). These feature vectors have been shown to have a linear relation with measured brain responses. However, this approach entails information loss since the target images need to be reconstructed from the linear prediction using an approximate inversion network such as a variational decoder, leading to a severe bottleneck to the maximum possible reconstruction quality.

In this paper, we avoid this sub-optimality by presenting to the participants photorealistic synthetic images generated using PGGAN. This allows us to store the ground-truth latents corresponding to the generated images which can be perfectly reconstructed using the generative model after predicting them from brain data.

## 2.2 NEURAL DECODING

### 2.2.1 PREDICTING LATENT VECTORS FROM BRAIN DATA.

We adapted the deep generative network of PGGAN by adding a dense layer at the beginning to transform brain data into latent vectors. This layer is trained by minimizing the Euclidean distance between true and predicted latent representations ($batchsize = 30$, $lr = 0.00001$, Adam optimization) with weight decay ($alpha = 0.01$) to reduce complexity and multicollinearity of the model. The remainder of the generative network was kept fixed.

The first decoding model is trained with a loss function that takes only the latent vectors into account. Yet, the ultimate goal is to reconstruct what our participants were seeing. Technically, this is achieved when all the layer activations up until the output image would be similar between the real and predicted latent vector. Therefore, we created five additional loss functions that include these layer activations to examine how these contribute to further optimization of neural decoding. Importantly, we only took the centers of the activation maps to exclude surrounding background noise. In the end, we trained one model on the latent vectors alone, and five models on the latent vectors together with one PGGAN layer activation.

### 2.2.2 PREDICTING LAYER ACTIVATIONS FROM BRAIN DATA.

Earlier work has found correspondences between artificial neural networks and the brain (7). Based on this knowledge, we trained four decoding models to predict PGGAN's layer activations from brain data to explore the correspondence between this deep generative network and the brain. Specifically, we used the following layer outputs of PGGAN: 4, 9, 14, and 19, to which we will refer to as layer activation 1, 2, 3, and 4, respectively, for the remainder of this manuscript. The loss function was the Euclidean distance between true and predicted layer activations (and not the latent vectors). The rest of training proceeded as before. Next, we examined the contribution of each voxel in a predefined mask on model performance using a combination of a searchlight mapping approach and occlusion analysis. The searchlight approach takes a cubic subset of $7 \times 7 \times 7$ voxels, centered on a voxel. As each voxel is $2 \times 2 \times 2$ mm$^3$, this results in volumes of $14 \times 14 \times 14$ mm $^3$. Neighboring voxels are only included when they are also in the mask. Ultimately, this searchlight is excluded from the brain data input to detect the effects of the center voxel on model performance.

## 2.3 DATASETS

### 2.3.1 VISUAL STIMULI

High-resolution face images ($1024 \times 1024$ pixels) are generated by the generator network of a Progressive GAN (PGGAN) model (11) from randomly sampled latent vectors. Each generated face image is cropped and resized to $224 \times 224$ pixels. In total, $1050$ unique faces are presented once for the training set, and 36 faces are repeated 14 times for the test set of which the average brain response is taken. This ensured that the training set covers a large stimulus space to fit a general face model, whereas the voxel responses from the test set contain less noise and higher statistical power.

### 2.3.2 BRAIN RESPONSES

An fMRI dataset was collected, consisting of BOLD responses that correspond to the perceived face stimuli. The BOLD responses (TR = 1.5 s, voxel size = $2 \times 2 \times 2$ mm$^3$, whole-brain coverage) of two healthy subjects were measured (S1: 30-year old male; S2: 32-year old male) while they were fixating on a target ($0.6 \times 0.6$ degrees) (19) superimposed on the stimuli ($15 \times 15$ degrees) to minimize involuntary eye movements.

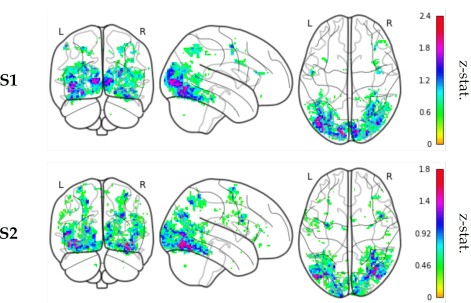

During preprocessing, the obtained brain volumes are realigned to the first functional scan and the mean functional scan, respectively, after which the volumes are normalized to MNI space. A general linear model is fit to deconvolve task-related neural activation with the canonical hemodynamic response function (HRF). Next, for each voxel, we computed its t-statistic and

Figure 3: Voxel mask: 4096 most active voxels based on highest z-statistics within the averaged z-map from the training set responses, resulting in a distributed network of activity.

converted these t-scores to z-statistics to obtain a brain map in terms of z per perceived stimulus. Ultimately, most-active 4096 voxels were selected from the training set to define a voxel mask (Figure 3). Most of these mask voxels are located in the downstream brain regions. Voxel responses from the test set are not used to create the voxel mask to avoid double-dipping.

The experiment was approved by the local ethics committee (CMO Regio Arnhem-Nijmegen). Subjects provided written informed consent in accordance with the Declaration of Helsinki. The fMRI dataset for both subjects and used models are openly accessible via Github.

## 2.4 EVALUATION

Model performance is assessed in terms of three metrics: latent similarity, feature similarity, and structural similarity. First, latent similarity is the Euclidean similarity between predicted and true latent vectors. Second, feature similarity is the Euclidean similarity between feature extraction layer outputs ($n = 2048$) of the ResNet50 model, pretrained for face recognition, which we feed stimuli and reconstructions. Lastly, structural similarity is used to measure the spatial interdependence between pixels of stimuli and reconstructions (22).

Next, based on the assumption that there exists a hyperplane in latent space for binary semantic attributes (e.g. male vs. female), (18) have identified the decision boundaries for five semantic face attributes in PGGAN's latent space: gender, age, the presence of eyeglasses, smile, and pose, by training five independent linear support vector machines (SVMs) (Figure 4). We used these decision boundaries to compute feature scores per image, by taking the dot product between latent representation and decision boundary (resulting in a scalar). In this way, model performance with regard to specific visual features could be captured along a continuous spectrum, and compared across images.

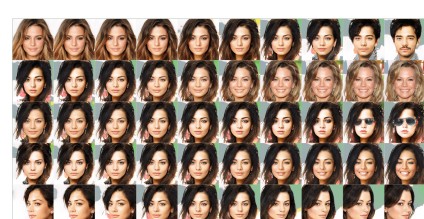

Figure 4: Semantic face editing of a stimulus image from the testing set (number 7) by varying the latent vector along a separation boundary.

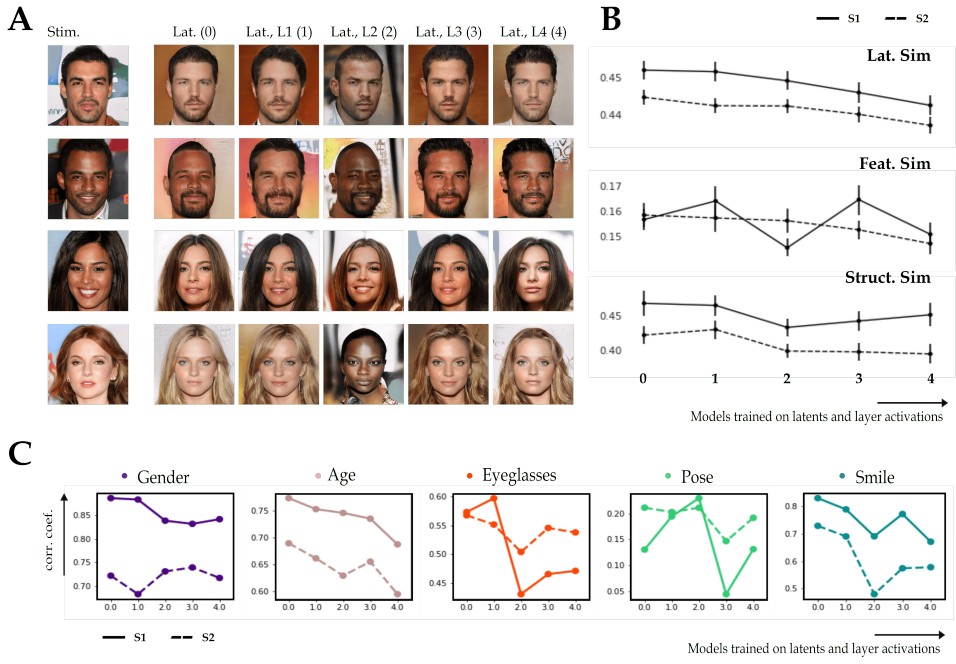

Figure 5: Results of the five models that predict latent vectors from brain data. The highest performance is achieved by our first model which is trained on latent representations alone and no intermediate layer activations. **A.** Four testset stimuli (left) and their five corresponding model reconstructions. **B.** The average latent similarity, feature similarity, and structural similarity (Y axis) of the five models (X axis). **C.** Five graphs display the Pearson correlation coefficients (Y axis) between true and predicted semantic feature scores of the five models (X axis) for each visual feature. We found high correlations for gender, pose, and age, but no significant correlation for the smile attribute.

## 3 RESULTS

Linear decoding of fMRI recordings using PGGAN's latent space has led to unprecedented stimulus reconstructions. The highest performance is achieved by the first model that is trained on latent vectors alone (Figure 5).

Figure 6 presents all image reconstructions of this best brain decoding model together with the originally perceived stimuli. To keep the presentation concise, the first half of the images (1-18) are reconstructed from brain activations from Subject 1 and the second half (19-36) from Subject 2. The interpolations visualize the distance between predicted and true latent representations that underlie the (re)generated faces. It demonstrates which features are being retained or change. The bar graphs next to the perceived and reconstructed images show the scores of each image in terms of five semantic face attributes in PGGAN's latent space: gender, age, the presence of eyeglasses, smile, and pose. Looking at the similarities and differences in the graphs for perceived and reconstructed images is a way to evaluate how well each semantic attribute is captured by our model. For most reconstructions, the two graphs match in terms of directionality. There are a few cases, however, demonstrating that there is still room for improvement, e.g. number 31, 34, and 35. Correlating the feature scores for stimuli and reconstructions resulted in significant ($p < 0.05$; Student's t-test) results for gender, age, eyeglasses, and pose, but not for smile (Figure 5). We would like to point out that using feature scores quantifies model performance as continuous rather than binary, explaining the significant correlation for eyeglasses despite lack of reconstruction in number 1 and 8.

Next, we compared the performance of the HYPER framework to the state-of-the-art VAE-GAN approach (21) and the traditional eigenface approach (3) which map the brain recordings onto different latent spaces. For a fair comparison, we used the same voxel mask to evaluate all the methods presented in this study without any optimization to a particular decoding approach. The

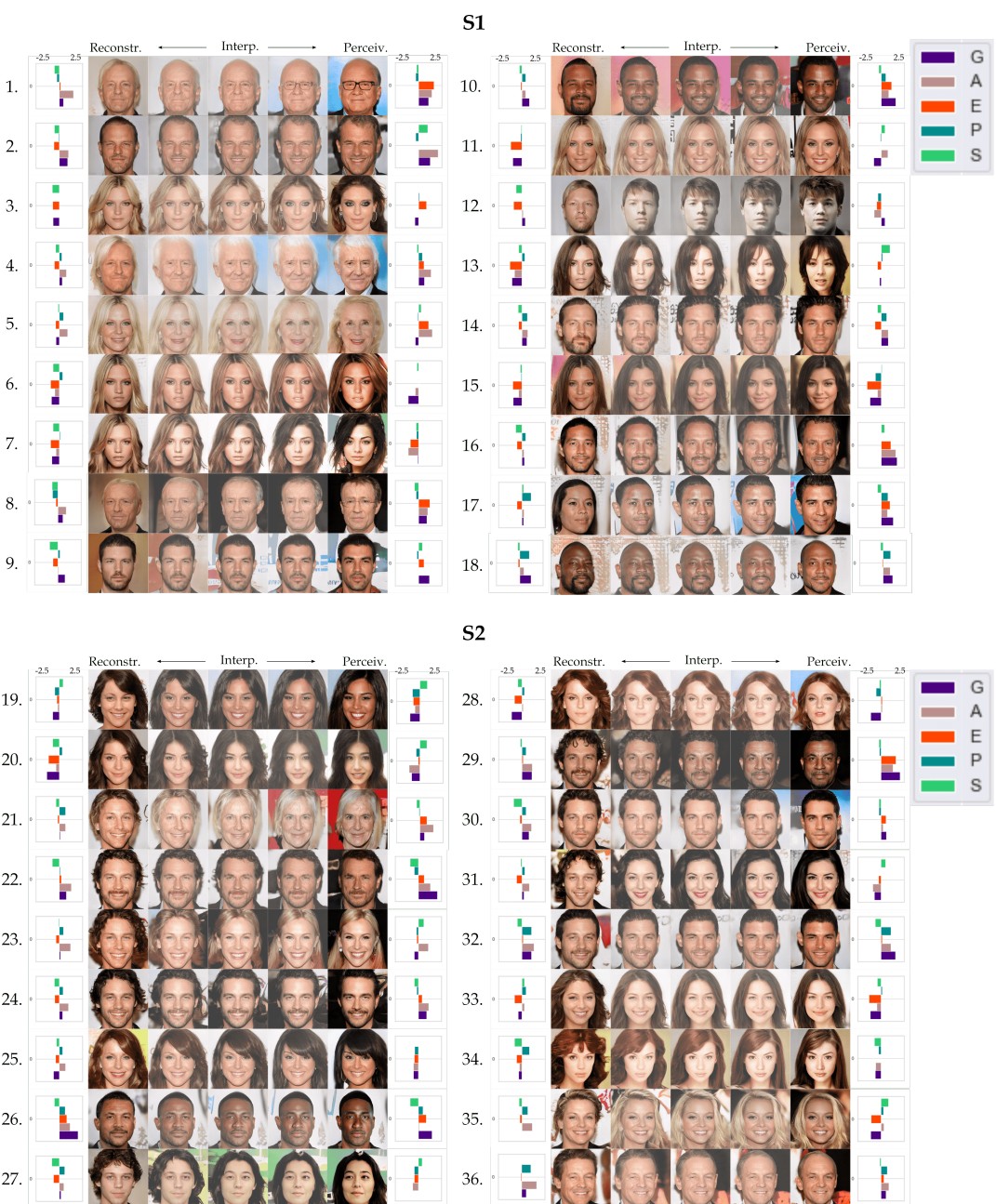

Figure 6: Results of model 0 that is trained on only the latent vectors. Here, we display testing set samples 1-18 for Subject 1 and 19-36 for Subject 2. Image reconstructions (left) versus perceived images (right). Interpolations visualize similarity regarding the underlying latent representations. Next to each reconstruction and perceived stimulus, a rotated bar graph displays the corresponding feature scores for gender, age, eyeglasses, pose, and smile.

VAE-GAN approach predicts 1024-dimensional latent representations which are fed to the VAE's decoder network for stimulus reconstruction ($128 \times 128$ pixels). The eigenface approach predicts the first 512 principal components (or 'eigenfaces') after which stimulus reconstruction ($64 \times 64$ pixels) is achieved by applying an inverse PCA transform. All quantitative and qualitative comparisons showed that the HYPER framework outperformed the baselines and had significantly above-chance latent and feature reconstruction performance (p « 0.001, permutation test), indicating the probability that a random latent vector or image would be more similar to the original stimulus (Table 1).

Table 1: Model performance of our method compared to the state-of-the-art VAE-GAN (21) and the eigenface approach (3) is assessed in terms of the feature similarity (column 2) and structural similarity (column 3) between stimuli and reconstructions (mean $\pm$ std error). The first column displays latent similarity which is only applicable to the HYPER method because the true and predicted latent vectors are known. Because of resolution differences, all images are resized to 224 $\times$ 224 pixels and smoothed with a Gaussian filter (kernel size = 3) for a fair comparison. In addition, statistical significance of the HYPER method is evaluated against randomly generated latent vectors and their reconstructions.

| | | Lat. sim. | Feat. sim. | Struct. sim. |
|---|---|---|---|---|
| **S1** | HYPER | $0.4521 \pm 0.0026$ | $0.1745 \pm 0.0038$ | $0.6663 \pm 0.0115$ |
| | | $(p < 0.001; perm.test)$ | $(p < 0.001; perm.test)$ | $(p < 0.001; perm.test)$ |
| | VAE-GAN | - | $0.1416 \pm 0.0025$ | $0.5598 \pm 0.0151$ |
| | Eigenface | - | $0.1319 \pm 0.0016$ | $0.5877 \pm 0.0115$ |
| **S2** | HYPER | $0.4447 \pm 0.0020$ | $0.1715 \pm 0.0049$ | $0.6035 \pm 0.0128$ |
| | | $(p < 0.001; perm.test)$ | $(p < 0.001; perm.test)$ | $(p < 0.001; perm.test)$ |
| | VAE-GAN | - | $0.1461 \pm 0.0022$ | $0.5832 \pm 0.0141$ |
| | Eigenface | - | $0.1261 \pm 0.0019$ | $0.5616 \pm 0.0097$ |

We also present arbitrarily chosen but representative reconstruction examples from the VAE-GAN and eigenface approach, again demonstrating that the HYPER framework resulted in markedly better reconstructions (Figure 7).

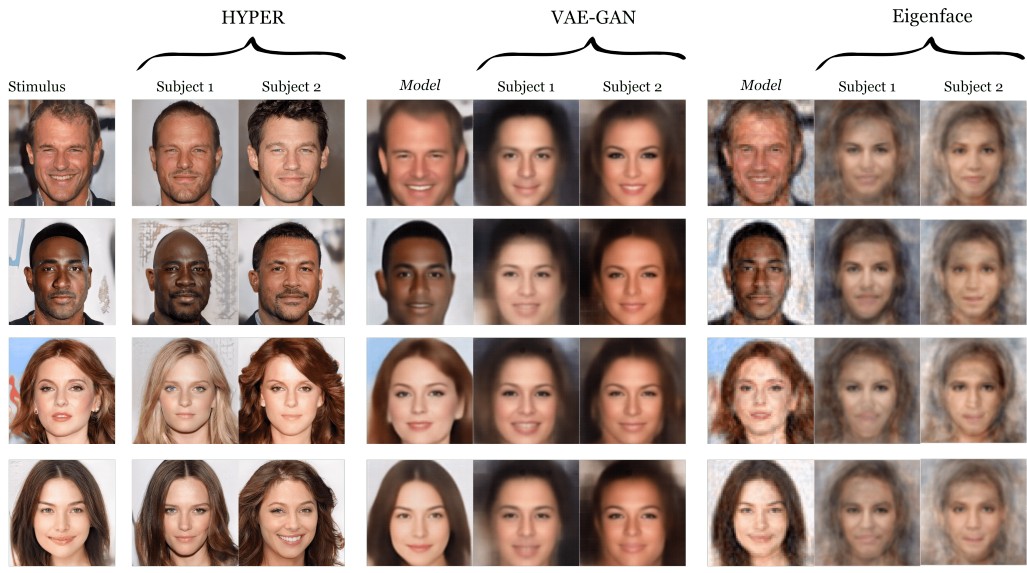

Figure 7: Qualitative results of our approach compared to (21) and the eigenface approach in reconstructing image 26, 28, and 36 (arbitrarily chosen). The *model* columns display the best possible results. For (21), this displays reconstructions directly decoded from the 1024-dimensional latent representation of this method. For the eigenfaces approach, this shows reconstructions directly obtained from the 512 principal components.

Lastly, we looked for any correspondence between the generative network and the brain. For each voxel, the searchlight-occlusion analysis identified on which of the four layer activation-predicting models it had the largest effect in terms of Euclidean activation similarity. This layer activation was then assigned to that voxel, and mapped to the brain (Figure 8). The majority of the voxels was found to correspond to layer activations closest to the latent vector (i.e., L1 and L2). Other than that, there was no systematic relationship between voxels and layer activations. This is not surprising considering that the voxel mask covered mostly the downstream regions such as FFA which are already specialized for high level representations. Nevertheless, it would be interesting to further investigate if GAN layers map onto the visual cortex similar to task-optimized models do when trained and tested on natural images.

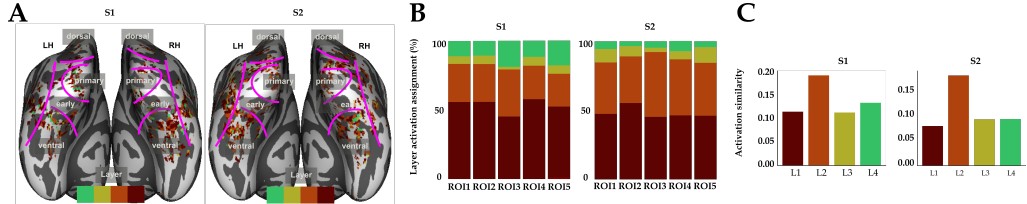

Figure 8: **A.** Layer activations that corresponded most with each voxel are mapped on the brain of two subjects respectively. In pink, borders are indicated between the primary visual cortex, the early visual cortex, the dorsal stream visual cortex, ventral stream visual cortex, and MT+ complex and neighboring visual areas, as based on (4). **B.** Distribution of layers assigned to voxels across different regions of interest. **C.** Activation similarity of different layers.

## 4 DISCUSSION

We have decoded brain recordings during perception of face photographs using the presented HYPER method, leading to state-of-the-art stimulus reconstructions. The HYPER framework resulted in considerably better reconstructions than the two benchmark approaches. It is important to note that the reconstructions by the VAE-GAN approach appear to be of lower quality than those presented in the original study. A likely explanation for this result could be that the number of training images in our dataset was not sufficient to effectively train their model (8000 vs 1050) and the different voxel selection procedure.

Image reconstructions by HYPER appear to contain biases. That is, the model predicts primarily latent representations corresponding to young, western-looking faces without eyeglasses because predictions tend to follow the image statistics of the (celebrity) training set. PGGAN's generator network is also known to suffer from this problem - referred to as "feature entanglement" - where manipulating one particular feature in latent space affects other features as well (18). For example, editing a latent vector to make the generated face wear eyeglasses simultaneously makes the face look older because of such biases in the training data. Feature entanglement obstructs the generator to map unfamiliar latent elements to their respective visual features. It is easy to foresee the complications for reconstructing images of existing faces.

A modified version of PGGAN, called StyleGAN (12; 13), is designed to overcome the feature entanglement problem. StyleGAN maps the entangled latent vector to an additional intermediate latent space - thereby reducing feature entanglement - which is then integrated into the generator network using adaptive instance normalization. This results in superior control over the semantic attributes in the reconstructed images and possibly the generator's competence to reconstruct unfamiliar features. Compared to PGGAN, the generated face photographs by StyleGAN have improved considerably in quality and variation, of which the latter is likely to elevate current biases. Replacing the PGGAN with StyleGAN would therefore be a logical next step for studies concerned with the neural decoding of faces.

Besides the large scientific potential, this research could also have societal impacts when enabling various applications in the field of neurotechnology (e.g. brain computer interfacing and neuroprosthetics) to help people with disabilities. While the current work focuses on decoding of sensory perception, extensions of our framework to imagery could make it a preferred means for communication for locked-in patients.

## 5 CONCLUSION

We have presented a framework for HYperrealistic reconstruction of PERception (HYPER) by neural decoding of brain responses via the GAN latent space, leading to unparalleled state-of-the-art stimulus reconstructions. Considering the speed of progress in the field of generative modeling, we believe that the HYPER framework that we have introduced in this study will likely result in even more impressive reconstructions of perception and possibly even imagery in the near future, ultimately also allowing to better understand mechanisms of human brain function.

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
