# OpenReview forum: "Hyperrealistic neural decoding: Reconstruction of face stimuli from fMRI measurements via the GAN latent space"
_ICLR.cc/2021/Conference — Reject_

### Official Review · AnonReviewer1 · 2020-10-13
**The stimuli were generated with the same GAN model**

**Rating:** 4
**Confidence:** 3

**Review:**

The paper proposes to reconstruct images of faces from fMRI measurements, using GANs. The authors collected a new dataset, showing static faces generated by a GAN model to human subjects, and recording their brain BOLD responses with MRI. Then, they learned a model to reconstruct the stimulus based on the brain responses. The authors demonstrate the high quality of the reconstructed faces, comparing with other recent methods.

Main limitations:

1. The reconstructed faces have a higher quality than previous work. However, the method is not conceptually different from e.g. [21], and the authors mostly used a better generative model, i.e. a better prior. In other words, since the generative model used (PGGAN) creates high-quality faces, they cannot reconstruct anything but high-quality faces. The quality of the faces is thus not a characteristic of the proposed method, but of the GAN model used. The authors should discuss this limitation in their paper.

2. More importantly, the authors generated all stimuli using the same GAN model as used in reconstruction. The space of faces is much larger than the space of faces generated by this particular GAN. This choice of stimulus makes the prior perfect, and other methods cannot compete fairly. To the extreme, one could use a GAN that always generates the same face, and decoding with this same GAN would then be perfect. This design choice makes any comparison with other methods unfair and thus impossible. To alleviate this major limitation, the authors could use a different GAN for reconstruction, specifically a GAN that was not trained on the same celebA dataset. However, it is not clear what training loss would be best in this case.

For these important reasons, I would rather recommend this paper for rejection.

Other proposed improvements:

3. In figure 6, I fail to understand how "interpolations visualize similarity regarding the underlying latent representations". Indeed, one can interpolate between any couple of latent representations, leading to a smooth interpolation between the faces. However, it does not give any clear insight about the similarity of latent representations. In fact, a lot of couples of faces seem rather far perceptually, such as 3, 5, 7, 12, 13, 16, 17, 21, 25, 27, 29, 31, 34, 35. If anything, the interpolations make the comparison more difficult. I would suggest removing them.

3. For reproducibility, the authors should mention how the latent space was sampled during the creation of the stimulus.

5. "elegantly" is not adding much to the paper.

6. "searchlight mapping approach and occlusion analysis" is not described in detail, nor linked to academic references.

7. In Figure 4, rows are not labeled.

8. In Figure 8, the legend should be ordered with the "smile" label at the top and the "gender" label at the bottom.

9. In the discussion, the mention of biases due to the particular distribution of PGGAN is a plus, though the authors could discuss more in detail the potential social impact of having such biases.

---

### Official Review · AnonReviewer3 · 2020-10-28
**A clever engineering of the decoding problem, but what novel insights on face perception ?**

**Rating:** 5
**Confidence:** 4

**Review:**

The manuscript entitled "HYPERREALISTIC NEURAL DECODING :
RECONSTRUCTION OF FACE STIMULI FROM MEASUREMENTS VIA THE GAN LATENT
SPACE" presents a novel percept reconstruction method based on
functional neuroimaging. The approach consists in decoding brain
responses via the GAN latent space, and leads to state-of-the-art
stimulus reconstructions.  More precisely, it is shown to outperform
VAE-GAN and eigenfaces. Experiments focus on metrics well suited to
face space analysis.

A nice feature of the paper is that it rests on a relatively straightforward
idea: simply learn to predict latent representations and then let the
magic of GAN-based face generation work. The main novelty is to use
generated samples as fMRI space, leading to a better anchoring of the
decoder in latent space.  While the main concept of the paper is quite
interesting, the overall quality of the paper is not great, due to
basic statistical (sample size, simple validation) and conceptual
(absolute lack of neuroscientific perspective) issues. To summarize,
the paper presents a clever engineering of the decoding problem,
but does not bring novel insights.


Major comments
==============

* My first concern is about statistics The test set is reduced to 36
  samples, which is strikingly insufficient to obtain reliable
  estimates. Indeed, the accuracy of the results will be with an
  uncertainty of about 1/sqrt(36) = 1/6. Even worse, the study design
  prevents cross-validation and is based on simple validation.  This
  means that essentially nothing -beyond the fact that the model is
  "significantly better than chance"- could be concluded from the
  results. In particular, it is claimed that "the HYPER framework
  outperformed the baselines" but is the difference significant ?
  Experiments are carried out on 2 subjects only, which is below the
  standards of the field. This means that the results may simply
  represent overfit.

* My second concern is that the model assumes that the brain is, so to
  say, constantly viewing faces: it says nothing about whether the
  subject is involved in face perception or not. While this sounds
  like a minor detail, it illustrates how narrow the perspective taken
  in the paper is: it is always possible to input brain activity to
  whatever-generator, and produce nice "hyperrealistic"
  hallucinations.  More deeply, this means that all this framework boils down to
   an engineering trick that does not inform us about the
  biological organization of perception.

* The embarrassing lack of neurophysiological relevance of the
  framework appears clearly in the following problematic statement
  "These feature vectors have been shown to have a linear relation
  with measured brain responses." this is problematic, because the
  latent space used by Gans or VAEs is somewhat arbitrary /
  unconstrained (even in terms of dimensionality). How comes
  that it can fit neural structures ? Nothing in the paper backs up
  the statement --- that is kept as vague as possible. Not
  surprisingly, biological relevance or impact are deferred to future
  work...

* The bibliography of the document is biased toward a certain
  lab, and thus unfair with respect to concurrent efforts, e.g.

R. Beliy, G. Gaziv, A. Hoogi, F. Strappini, T. Golan, and M. Irani,
“From voxels to pixels and back: Self-supervision in natural-image
reconstruction from fMRI,” in Advances in Neural Information
Processing Systems, 2019.

Reconstructing faces from fMRI patterns using deep generative neural networks
R VanRullen, L Reddy. Communications biology 2 (1), 193

Transfer learning of deep neural network representations for fMRI decoding
M Svanera, M Savardi, S Benini, A Signoroni, G Raz, T Hendler, L Muckli, ...
Journal of neuroscience methods 328, 108319

In the discussion section at least, it seems necessary to report about other efforts on handwritten digits decoding, e.g.

Reconstructing imagined letters from early visual cortex reveals tight topographic correspondence between visual mental imagery and perception
M Senden, TC Emmerling, R Van Hoof, MA Frost, R Goebel
Brain Structure and Function 224 (3), 1167-1183


Minor comments
==============

• Method presentation is not extremely clear: from the methods section
  I thought that a searchlight approach was used, which begs the
  question of how a global decoding performance is obtained. My
  understanding is that both searchlight- and whole brain analyses are
  carried out, but this is not clear.

• The authors should indicate which tool is used to perform their
  analyses (data processing; visualization etc.)

• "We found high correlations for gender, pose, and age, but no
  significant correlation for the smile attribute." is weird to me
  because pose has the lowest correlations. Maybe I am missing
  something here.

---

### Official Review · AnonReviewer4 · 2020-10-29
**Valuable work contributing to the interdisciplinary field of computer science and cognitive neuroscience**

**Rating:** 7
**Confidence:** 5

**Review:**

The manuscript entitled “Hyperrealistic neural decoding: Linear reconstruction of face stimuli from fMRI measurements via the GAN latent space” utilizes a GAN-based network structure for generating faces that are presented to the subjects during fMRI acquisition. The acquired fMRI signals are then used to predict latent vector in GAN, where the predicted vectors (rather than the original, trained vectors) are used to generate “fMRI-derived” faces in turn. This work proposed a novel perspective to the field of cognitive neuroscience and human brain mapping in functional neuroimaging studies. It is an indirect yet highly effective way for modeling brain functional signals, and representing how the visual encoding-decoding actually works in human brain. Similar schemes of investigation can be easily developed based on the proposed work (e.g. analyzing correspondence between EMG and fMRI during motor tasks), which will bring significant value to the community.

This work has great potentials, yet several interesting and important points of investigation were skipped in the study design, which lowered the scientific value of it and the validity of the model. Specifically,
1)	The framework needs to include more semantic information into face generation and analysis: currently only simple features including age, gender, eyeglasses, smile, and pose are considered. Even for these features, evaluation of the predicted latent features and corresponding reconstructed face images w.r.t. their semantic information, as covered in Fig. 5, is conflicting, insufficient, and difficult to interpret: Firstly Fig. 5(c) implies low correlation between the true/predicted semantic scores for “Pose”, but in the caption and text it is stated that “but no significant correlation for the smile attribute”. Secondly, group-wise Pearson correlation score for the semantic features can only support the accuracy of the generated latent features, which has already been illustrated in Fig. 5(b) and Table 1. The reviewer suggests that individual semantic response (in the form of reconstructed faces) to the face stimulus might be more interesting to investigate. Finally, it is counter-intuitive to find that the first (trained on latent representations alone) can achieve the best performance even for semantic feature prediction, as image semantics such as pose and glasses are well-represented on the image itself.
2)	Analysis of brain activation map based on the learned model, as shown in in Fig. 8, is difficult to read and interpret. ROI1~5 in Fig. 8 (b) cannot find the corresponding definitions for their names (is ROI1 visual cortex?). Implication of Fig. 8 (c) is difficult to understand and not provided in the text.

---

### Official Review · AnonReviewer2 · 2020-10-29
**More accurate reconstruction of stimulus from fmri brain response**

**Rating:** 5
**Confidence:** 4

**Review:**

This work concerns reconstruction of visual stimulus from brain state as measured by fmri.

This is a quite well-explored field with an already impressive state of the art based on deep network methods. State of the art is well described in the paper and is represented e.g. by architectures, similar to the pipeline developed for the present study.
The main motivation of the present paper is to invoke a more complex image reconstruction network (GAN based).
The improved visual quality is quantified in several metrics (including the ability to reconstruct / represent face relevant features).

Reservations: The novelty is limited compared to the existing baselines mentioned in the submission.
 While visual fidelity / quality is a basic important dimension to "brain read out", the quality presented for the competing baseline seems lesser that in the original baseline paper(s), hence questioning these papers. However, I can not rule out that the present study doesn't get the baseline correct (one potential reason being data size as mention in the submission).

Conclusion: The novelty is limited and the author miss the opportunity to ask novel research questions. The comparison with a baseline is hampered by lower visual performance of the present instantiation than found in the original paper.
In a sense the author have missed the opportunity to ask novel research questions in this interesting domain. It is unclear (to me) if the improved visual quality is simple due to the improved performance of the GAN or the ability to read out more relevant features from the fmri.

---

### Official Review · AnonReviewer5 · 2020-11-04
**Low significance and ethical concerns**

**Rating:** 2
**Confidence:** 5

**Review:**

Short summary
---------------------
The authors propose a framework combining a GAN and linear encoder and decoders to reconstruct perceived face stimuli from fMRI data. They compare their framework to two baselines in the field and display a higher similarity (in different spaces) between the reconstructed images of their method and the stimuli, compared to the baselines.

Strengths
--------------
The framework is simple and allows for using different models to generate the stimuli, encode brain activity, decode brain activity and decode the stimuli. Appropriate baselines are selected and the authors quantify the similarity between generated and reconstructed images in different spaces.

Weaknesses
-------------------
I am uncertain of the impact of the proposed approach, as it does not propose techniques to investigate brain functioning, nor does it provide a means of communication with disabled patients (as claimed by the authors). I see Ethical concerns with this type of model, which, in my opinion, are not counterbalanced by its usefulness.

Novelty
-------------
The approach combines well established models and techniques into a novel framework. While I found some creativity in the setup, the novelty is overall low and I am not convinced that some of the techniques used are not acting as bottlenecks (see detailed comments).

Clarity
-----------
The paper is relatively clear. I appreciated the presence of multiple figures and of various examples. I believe that the methods could be clearer (see detailed comments).

Significance
----------------
To me this was a major concern with this work as I found some claims bold and not substantiated by the experimental setup or the results. For instance, the authors claim to decode naturalistic stimuli. However, they can decode GAN generated images, which is substantially different, especially given the approach to average the fMRI signals over 14 repetitions in the test set to increase SNR.

Rigor
--------
Overall, I found that the work was relatively well performed, although not excellent. I wished that the comparison with the VAE approach was fair and would suggest that the authors work towards achieving SOTA in their baselines for a fair comparison.

Detailed comments
---------------------------
- Faces generated by a GAN cannot be deemed naturalistic
- Isn’t the setup circular? How would that model help understand face processing in naturalistic settings?
- Ethics concerns: how is the application helping people with disabilities or understanding brain function? The authors mention this as a potential communication means for locked-in patients. However, (1) the results are not strong enough to suggest a potential communication tool, (2) the reliance on fMRI signals makes this impractical and expensive. There are no conclusions regarding brain function that this technique brings, especially given the voxel selection based on a linear regression model.
- Novelty and technical sophistication is rather low: combining existing techniques in a novel system.
- A better test would have been to reconstruct stimuli that were not generated by the GAN
- “Importantly, we only took the centers of the activation maps to exclude surrounding background noise”. I believe the authors refer to “activation maps” as the fMRI z-score maps. This can however be confusing for an ML reader. Please clarify the language.
- It is unclear to me what the goal of the “five additional loss functions” is, or how they are formulated.
- why is the test set not sampled the same way as the training set? While taking the average of 14 repetitions increases signal-to-noise ratio, this setting further departs from any naturalistic decoding.
- An fMRI study with 2 subjects is not representative, given inter-subject variability. This is further compounded by the low number of test images (36 per subject after averaging).
- There is a clear imbalance in the generated stimuli in terms of age, race or whether they wear glasses. This limits the impact of the scores for the 5 different attributes. It is also a reason why neuroscience experimental stimuli are thoroughly controlled for. While this is touched upon in the discussion, this could be an indication that the proposed approach would fail on naturalistic stimuli.
- Some reconstructed stimuli are highly similar, despite different generated images (e.g. 23 and 24). What could explain this phenomenon? Could there be some type of mode collapse in the reconstruction? It would be interesting to compute the pair-wise similarity between reconstructed images compared to the pairwise similarity of the generated images (in the different spaces mentioned, i.e. latent, feature, attributes).
- Isn’t the linear regression model a bottleneck here? Why not use the “raw” BOLD signal instead of z-scores? This reflects an assumption that the encoding between stimulus and activation map is linear. Couldn’t there be a non-linear mapping between stimulus perception and the latent space? Overall, the proposed framework relies on established techniques without questioning or reflecting on their assumption.

Minor
-------
- Given the linear model used, why limit to 4096 voxels? This seems like an arbitrary number. Is it related to the z-scores or a specific p-value threshold on the activation map?
- The relationship between the trained models and the ResNet mentioned in section 2.4 is unclear. Is it used for evaluation as another way to estimate the similarity between the reconstruction and the GAN generated images? Is there a justification/reference for this technique?
- Figure 5C, could the authors use the same y-scale? The legend mentions “We found high correlations for gender, pose, and age, but no significant correlation for the smile attribute.” Pose however had the lowest correlation values. How about eyeglasses?
- “[…] permutation test), indicating the probability that a random latent vector or image would be more similar to the original stimulus”. This sentence is unclear to me: is the permutation test assessing whether HYPER has significantly higher similarity between the reconstructed image and the generated image than if using a random latent vector? Please provide more details on the hypothesis tested by the permutation test in each space, as well as how these tests compare the different techniques. Couldn’t the permutation tests be applied to the baselines techniques as well?

---

### Decision · Program_Chairs · 2021-01-07
**Final Decision**

**Decision:**

Reject

**Comment:**

The approach proposed here have raised major concerns from multiple reviewers especially concerning the novelty and the experimental validation procedure.